# The Image of Jews According to the Canonical Tradition of the Orthodox Church

**Răzvan Perșa**

Faculty of Orthodox Theology, Babeș-Bolyai University, 400084 Cluj-Napoca, Romania; razvan.persa@ubbcluj.ro

**Abstract:** In this study, the author attempts to analyze the canons of the Orthodox Church, which are still normative for all Orthodox Autocephalous Churches, in order to identify the image of Jews and the restrictions or prohibitions imposed by these canonical texts. The paper is structured in three parts that analyze the canonical provisions regarding Jewish religious influences, Judaizing practices, Jewish–Christian religious syncretism, and social interaction with Jews based on religious influences. The main purpose of the present paper is to identify whether the Canonical Tradition of the Orthodox Church contains a form of anti-Semitism or exclusion of Jews on the basis of these texts. A well-articulated contextual interpretation of the canons is necessary to understand the relationships between Christians and Jews in the Byzantine period. To this end, the canons of the Orthodox Church will be interpreted in the social, historical, doctrinal, and canonical context of their promulgation in the life of the Orthodox Church.

**Keywords:** Jews; canon law; anti-Semitism; Orthodox tradition; canons

## 1. Introduction

In the context of the manifest exaltation of Nazi and fascist ideologies in Romania in the inter-war period, the Romanian legislature promulgated certain discriminatory laws against Jews, first, on 22 January 1938, the law on the revision of Romanian citizenship, then, on 8 August 1940, the law on the legal status of the Jewish inhabitants of Romania. In other words, a clear delimitation on the criterion of blood was made between citizens of Romanian nationality and Jews, even those who converted to Christianity, with all Jews being considered second-class citizens. Some Orthodox theologians considered that the conversions of Jews were merely shallow, and they brought to bear certain canons issued in the Byzantine period to support this attitude. Unfortunately, their interpretation of these canons turned into an anti-Semitic interpretation, serving simply as a pretext to support their fascist or Nazi ideas and to camouflage their racial hatred. To achieve this, a decontextualized interpretation of the canons of the Church was used in their argumentation (Racoveanu 1940, pp. 20–21; Stan 1941, pp. 1–2; Stan 1942, p. 144), which had severe consequences for the Jews in Romania (Petcu 2018, pp. 117–18).

Conversely, a decontextualized interpretation of the canons of the Church arose following the tragic events of the Second World War, which persists to this day. Many scholars consider these canons not just texts against Jewish religious or cultic influences or Jewish proselytism, but as anti-Jewish, and even anti-Semitic texts, portraying a discourse of hate, intolerance, ethnic hatred, and anti-Semitism, starting from the first Christian centuries. According to this interpretation, a rewriting of history on the basis of this kind of interpretation is a desideratum. In the epilogue of his book, Robert Wilken said:

> "Every act of historical understanding is an act of empathy. When I began to study John Chrysostom's writings on the Jews, I was inclined to judge what he said in light of the unhappy history of Jewish–Christian relations and the sad events in Jewish history in modern times. As much as I feel a deep sense of moral responsibility for the attitudes and actions of Christians toward the Jews, I am no

longer ready to project these later attitudes unto the events of the fourth century. No matter how outraged Christians feel over the Christian record of dealing with the Jews, we have no license to judge the distant past on the basis of our present perceptions of events of more recent times". (Wilken 1983, pp. 162–63)

For this reason, a well-articulated contextual interpretation of the canons is necessary to understand the relationships between Christians and Jews in the Byzantine period.

*Materials and Methods*

The initial research material for the present study consisted of the Collection of Canons of the Orthodox Church, approximately 770 canons (Wagschal 2015, pp. 60–61), which are still considered normative for the ecclesial organization and church discipline by all autocephalous Orthodox Churches. The research material did not include non-canonical texts from collections that were not recognized by the Eastern Orthodox Church in her Corpus Canonum (Troianos 2012, pp. 115–24), due to their local usage and non-obligatory canonical status. The first stage of the research was to identify from these 770 canons those texts that imposed, as we will see, a number of restrictions or prohibitions on Christians regarding Jewish religious influences, Judaizing Church practices and Jewish–Christian religious syncretism, and social and religious interaction with Jews. The main material used for the present research consisted of 21 canons from the Canonical Collection of the Orthodox Church. This material represents all the canons from the Corpus Canonum regarding Jews. The selected material was grouped into three categories, to which I have added subcategories. The main purpose of this thematic content analysis was to identify thematic categories of canonical restrictions regarding Jews in order to compare them with thematic categories of anti-Semitism. The second step is to see the presence or absence of certain categories by applying quantitative content analysis. From the start, it is vitally important to emphasize the difference between, on the one hand, the legal provisions regarding Jews in the Byzantine period (Linder 2006, pp. 128–69), the so-called Jewry Law (Linder 2012, pp. 195–206) that had a direct influence on the legal status of Jews in society, and, on the other hand, Canon Law that applied only to members of the Church: clergymen, laymen, and monastics. Accordingly, the canons found in the Canonical Collection only indirectly influenced the status of Jews in the Byzantine Empire. This research uses content analysis as the main method applied to the canonical material, based on the principles of the interpretation of the canons and legal texts in the Orthodox Church (Perșa 2021, pp. 445–547). The selected canonical material was analyzed using both quantitative and qualitative methods, based on current research on religious studies (Brink 1995, pp. 461–75) and legal or canonical texts (Stanisz et al. 2022, p. 5). From the perspective of qualitative content analysis, the canonical material was subjected to synchronic and diachronic interpretation to emphasize the reception history, or Wirkungsgeschichte, of these texts in Orthodox Church Tradition.

In order to see if the Corpus Canonum contains forms of anti-Semitism or exclusion of Jews based on these canonical texts, a working definition of anti-Semitism is needed. The difficulty of such a definition is that no attempt to define anti-Semitism has been unanimously accepted by current academic research. Some scholars impose a difference between anti-Semitism, as racial discrimination towards Jews in modern times, and anti-Judaism, as a form of rational, nonrational, and irrational response to Jewish religious influences (G. Langmuir 1990, pp. 57–99). The distinction was criticized by many scholars (Schäfer 1997, pp. 197–211), but this pioneered direction was developed by further academic research, as it was emphasized by Robert Chazan in his recent book (Chazan 2016, pp. viii–xi). A more elaborated definition was adopted by the International Holocaust Remembrance Alliance in 2016 based on previous debates (IHRA 2016, p. 1). This definition was endorsed by many countries, governmental bodies, and the European Union and the Council of Europe (Weitzman 2019, pp. 463–71). According to the IHRA document: "Antisemitism is a certain perception of Jews, which may be expressed as hatred toward Jews. Rhetorical and physical manifestations of anti-Semitism are directed toward Jewish or non-Jewish

individuals and/or their property, toward Jewish community institutions and religious facilities." (IHRA 2016, p. 1). According to this definition, the present research attempts to see if the canons of the Orthodox Church are expressions of hatred toward Jews, or if they are rhetorical and physical manifestations directed toward Jews or their property, or Jewish community institutions and their religious facilities.

## 2. Canons Regarding the Prohibition of Jewish Religious Influences on Christians

### 2.1. Jewish Passover and Christian Pascha

Some scholars may consider that the canons regarding the date of Christian Pascha represent internal problems of the Church and that they do not refer to the relationship between Jews and Christians. Nonetheless, by analyzing these canons, we can observe what kind of influence the Jewish community had on Christians in the first centuries, especially regarding the single most important feast of Christianity: Pascha.

As Pascha is one of the fundamental moments of Christian liturgical life, the canons totally prohibit any influences of Jewish Passover on Christian Pascha. These prohibitions have to be understood in the broader doctrinal context of the fourth century, which was dominated by Arianism and other heresies against the divinity of Christ. In this context, the canons often refer to three categories: Jews, pagans, and heretics, all of whom were accused of denying the divinity of Christ. The same distinction can be found in Western Canon Law (Freidenreich 2013, pp. 73–91). We have here an exclusion based on doctrinal rather than racial differences.

The Canonical Collection of the Orthodox Church begins with the so-called Apostolic Canons (Schwartz 1963, pp. 274–362), a collection of 85 canons, part of Book VIII, Chapter 47 of the Apostolic Constitutions (Διαταγαί τῶνἁγίων Ἀποστόλων, Constitutiones Sanctorum Apostolorum). The Apostolic Canons represent a canonical compilation of Syriac origins (Metzger 1985, pp. 275–309). Most probably they were compiled in Antioch, in the second part of the 4th century (Joannou 1962, p. 1; Gaudemet 1958, p. 45), between 375 and 380, from several liturgical, canonical, and theological texts from previous centuries (Synek 1997, p. 1; Metzger 1985, pp. 58–59; Schwartz 1963, p. 12; Turner 1930, pp. 128–41). These canons are very important for our debate because they present the canonical norms applied in Antioch at the end of the 4th century. Their interpretation is fundamental for a contextual understanding of the work of John Chrysostom regarding Jews and the Jewish problem in Antioch (Hosang 2010, pp. 109–12).

The first canon of the so-called Canonical Collection of the Orthodox Church that refers to Jewish religious or cultic influences on Christians is the 7th Apostolic canon, regarding the timing of the Christian Pascha is relation to the Jewish Passover. It should be noted in passing that, despite dedicating an entire chapter to "Jewry-Law in Byzantine Canon Law" in his paper, Amnon Linder omitted this canon, although it is one of the most important texts regarding prohibitions of Jewish influences on Christians. His chapter is more of a presentation of Byzantine Canon Law than a commentary. His interpretation of the canons is more focused on the understanding of the Byzantine canonists, Zonaras, Balsamon, and Aristenos, than on the interpretation of the canons in their context (Linder 2006, pp. 195–206). Apostolic Canon 7 states that:

> "ζ. Εἴ τις ἐπίσκοπος, ἢ πρεσβύτερος, ἢ διάκονος, τὴν ἁγίαν τοῦ Πάσχα ἡμέραν πρὸ τῆς ἐαρινῆς ἰσημερίας μετὰ Ἰουδαίων ἐπιτελέσοι, καθαιρείσθω" (Perșa 2022, p. 105). "If any Bishop, or Presbyter, or Deacon celebrate the holy day of Easter before the vernal equinox with the Jews, let him be deposed". (Cummings 1957, p. 9)

The same prohibition can be found in the first canon of the Council of Antioch (Stephens 2015, p. 241). This canon shows the immediate necessity of solving such an important problem at that time (Feldman 1996, p. 399). We can observe that the canon speaks about the decision of the Fathers of the First Ecumenical Council. Despite the fact that the common date of Easter and its calculation were important issues at the beginning of the fourth century for all of Christendom, the Fathers of the Council of Nicaea, where

these problems were debated and decided, did not issue any canon or rule that could be included in the canons of the Council. Unfortunately, no document from the First Ecumenical Council regarding the criteria for calculating the Easter date has survived. The "discovery" of such a document by J. B. Pitra (Pitra 1858, p. 541) has been dismissed by many scholars as uncertain or inauthentic (Schmid 1905, p. 66; Huber 1969, p. 65), but considered authentic by others (de Urbina 1963, pp. 106, 228; Larentzakis 1979, p. 68). The only firm canonical rules regarding the calculation of the Easter date are the 7th Apostolic Canon and the first canon of the Council of Antioch. We can also find some patristic references regarding this issue (L'Huillier 1996, pp. 19–30; Mali 2004, pp. 309–327).

What can be observed from the beginning in both canons is the fact that they prohibit the celebration of Easter "with the Jews (μετὰ τῶν Ἰουδαίων)". How can the phrase "μετὰ τῶν Ἰουδαίων" be understood? Does it mean that the Christians should not celebrate the Jewish Passover on the same date, i.e., 14 Nissan, because it was not always on a Sunday and the Jewish moon calendar was different from the Christian solar calendar (L'Huillier 1996, pp. 24–25; Ogitsky 1973, p. 276; Calivas 1990, p. 338), and in that period much altered? Or is it to be understood simply that, in the event that the Jewish Passover falls on the same date as the Christian Easter (i.e., on a Sunday), then the latter feast would need to be postponed, since celebrating the Christian Pascha on the same day as the Jewish Passover is totally prohibited?

These questions are of great importance, not just from the canonical perspective but also from liturgical, theological, and ecumenical perspectives. If we understand the phrase "μετὰ τῶν Ἰουδαίων" as a total prohibition of celebrating Easter with the Jews because of Jewish proselytism, influences, and their partially erroneous way of calculating the Passover, then Christians should not take into consideration the date of the Jewish Passover at all. But if we consider the phrase "μετὰ τῶν Ἰουδαίων" as a total prohibition of celebrating Easter on the same day as Jewish Passover when this occurs on a Sunday, then the Christian Easter will have to be postponed, and the Jewish Passover then becomes one of the criteria in calculating Christian Pascha. If, from a liturgical point of view, we consider the latter interpretation as being the correct one, then all Orthodox feasts and the entire liturgical calendar are totally dependent on the Jewish Passover. From an ecumenical point of view, as we can see, the result of the interpretation of this phrase will lead us to different dates for the celebration of Easter by Christian denominations, especially the difference between the Orthodox on one side and Roman Catholics and Protestants on the other.

The canonical and patristic tradition of the first millennium supports the first understanding of the canon as a total rejection of the date of Jewish Passover as a criterion in the computation of Christian Pascha. For example, Emperor Constantine, addressing the other local churches, says:

> "By rejecting their custom, we establish and hand down to succeeding ages one which is more reasonable, and which has been observed ever since the day of our Lord's sufferings. Let us, then, have nothing in common with the Jews, who are our adversaries. For we have received from our Saviour another way . . . What motive can we have for following those who are thus confessedly unsound and in dire error? For we could never tolerate celebrating the Passover twice in one year". (Theodoret of Cyprus 2019, p. 32).

St. Epiphanius of Salamis mentions that the three criteria decided by the Fathers of the First Ecumenical Council were: the vernal equinox; the full moon; and Sunday. There is no mention of the Jewish calculation criteria.

> "For the fixing of the date of the Paschal Feast is determined by three factors: from the course of the sun; because of the Lord's Day; and because of the lunar month which is found in the Law, so that the Passover may be held on the fourteenth of the month as the Law says. Thus it cannot be celebrated unless the day of the equinox is past, although the Jews do not observe this or care to keep so important a matter precise; with them, everything is worthless and erroneous" (Epiphanius 2013, p. 423).

The main point of St. Epiphanius is that the Jews can celebrate their Passover even before the vernal equinox. If the Christians follow the same reasoning, then they can celebrate Easter two times a year, because—as was the custom in Epiphanius' day—the new year starts after the vernal equinox (Epiphanius 2013, p. 424).

Being a part of the Apostolic Constitutions, the 7th Apostolic Canon refers to the same prohibition: not to celebrate the Christian Passover with the Jewish Pascha because the Jews take neither the vernal equinox nor the Sunday into account. Because of this, they could have celebrated the Pascha twice a year, on any day of the week. This explanation is given in the Apostolic Constitutions, Book V, chapter 17 (Donaldson 2007, p. 447). The prohibition of celebrating Easter with Jewish Pascha was reiterated by some of the Fathers of the Orthodox Church.

At the beginning of the second millennium, the interpretation of the canon developed into a total prohibition of celebrating with the Jews and a postponement of Christian Easter if this happened. According to the three criteria, if the calculation of the Jewish Pascha were correct, then, astronomically, 14 Nisan would have been 14 days after the New Moon, and accordingly, with a full moon. In this case, it would have been impossible for the Church to have the same date for Pascha as the Jewish Passover because Pascha would have been on the first Sunday after the full moon after the vernal equinox. Regarding the date of Easter, another issue arose, namely: the lack of precise mathematical accuracy of the Julian calendar as well as of the lunar calendar, both problems that persist in the current Orthodox Paschalion.

The first Byzantine author to interpret the 7th Apostolic Canon as a total interdiction of Christians celebrating Easter with the Jews by imposing a requirement to postpone Easter if both fall on the same date was Johannes Zonaras (Rhalles and Potles 1852, p. 10). By this, he imposed a total dependence on the Byzantine computation of the Easter date on the Jewish Pascha. He was followed by other Byzantine canonists, such as Theodor Balsamon (Mali 2004, p. 314). They extend the prohibition of celebrating Easter on 14 Nisan, not because it could be before the vernal equinox, or on any day of the week, as the patristic and canonical tradition of the first millennium attested, but because of the necessity of postponing Easter and the total dependence on the Jewish Passover (Calivas 1990, pp. 333–43). The problem in their interpretation is the fact that, in the 12th century, as Zonaras has said, nobody knew exactly when the vernal equinox occurred. Aristenos, the first to write short commentaries on the canons, does not even mention the vernal equinox. His interpretation is clear: "It is simple. He who celebrates the Passover together with the Jews is defrocked." (Rhalles and Potles 1852, p. 11) These Byzantine canonists were the greatest promoters of the dependence on the Jewish calendar (Ogitsky 1973, p. 278). The interpretation by the Byzantine canonists erred due to its being a decontextualized exegesis of the 7th Apostolic canon, influenced by the realities of the 12th century when the Orthodox Easter date was behind the astronomical date and the Jewish computation of Passover. The Jewish Paschalion became one of the criteria for computing the date of Christian Passover because, in that period, the Jews were calculating the first full moon more accurately.

In the 14th century, the Byzantine Church underlined the necessity of calendar reform. Thus, the philosopher and astronomer Nikephoros Gregoras proposed, in 1325, to Emperor Andronicus II Paleologos some adjustments to the calendar of that time (Calivas 1990, p. 339). The most important problem was considered to be the date of the vernal equinox. In 1335, the famous canonist and monk Mattew Blastares promoted the same need for calendar reform. In 1371, astronomer and monk Isaak Argyros proposed a change to the Paschalia, according to the astronomic spring equinox (Calivas 1990, p. 339). Unfortunately, none of these proposals for calendar reforms were followed through.

Matthew Blastares considers in his Synthagma that the date of Easter has to be computed according to four criteria: (1) after the vernal equinox; (2) not on the same day as the Jews; (3) after the first full moon; and (4) during the first week after the first full moon.

This interpretation was made by Nikodemus the Hagiorite as well, despite the fact that he considered the Julian calendar outdated and wrong in its computation. Following Saint Nikodemus the Haghiorite, 19th and 20th-century canonists and theologians considered that the 7th Apostolic Canon refers to Jewish Passover as one of the criteria for calculating the date of Easter, and it is a theological anti-canonical error to celebrate Easter on the same day with them.

In conclusion, Orthodox Easter has to be postponed if it is on the same date as the Jewish Passover. Although the purpose of the canon was to avoid any Jewish influence on the date of Easter in the Orthodox Church, the development of the interpretation of the canon (Wirkungsgeschichte) led to a real dependence of the Orthodox Churches on the date of the Jewish Passover.

### 2.2. Jewish Feasts in Christian Practice

Relating to the same problem, another Apostolic Canon, Canon 70, forbade any influence of Jewish feasts on Christian church life.

> "Εἴ τις ἐπίσκοπος, ἢ πρεσβύτερος, ἢ διάκονος, ἢ ὅλως τοῦ καταλόγου τῶν κληρικῶν, νηστεύοι μετὰ Ἰουδαίων, ἢ ἑορτάζοι μετ᾽ αὐτῶν, ἢ δέχοιτο παρ᾽ αὐτῶν τὰ τῆς ἑορτῆς ξένια, οἷον ἄζυμα ἤ τι τοιοῦτον, καθαιρείσθω εἰ δὲ λαϊκὸς εἴη, ἀφοριζέσθω" (Perșa 2022, p. 128). "If any bishop, presbyter, or deacon, or any one of the list of clergy, keeps fast or festival with the Jews, or receives from them any of the gifts of their feasts, as unleavened bread, any such things, let him be deposed. If he be a layman, let him be excommunicated"

(Percival 1900, p. 598).

The phrase "τὰ τῆς ἑορτῆς ξένια", "the gifts of the feast", refers to the unleavened breads or "ἄζυμα" of Passover. The canon tried to stop any Judaizing perception of fasting and feasting according to Jewish tradition. The canon addresses both upper and lower clergy, thus showing that even senior clergymen were influenced by Jewish liturgical practices, as were also laymen. This canon supplements the rule found in the 7th canon and is aimed at Christians who were celebrating the Jewish Passover and, after that, were keeping the days of fasting with unleavened bread. The difference between the Jewish Passover and Christian Pascha emphasized by the Father of the Church was that the Jews were celebrating by feasting and then fasting, i.e., keeping the days of unleavened bread, whereas Christians have to fast for a period and then may feast.

Regarding the Jewish feasts, Canon 37 of Laodicea decided: "That it is not permitted to accept festival gifts from Jews or heretics, nor to celebrate jointly with them." The word "ἑορταστικά" can be translated here as well as "festal gifts". Some scholars suggest that the canon is speaking about the Jewish feast of Purim (Schrechenberg 1990, p. 278). It is most probable that the canon refers to the Jewish Passover and to the gifts given at this feast (Hosang 2010, p. 101). As we saw, the same prohibition is stipulated in the 70th Apostolic Canon. Canon 38 of the Council of Laodicea adds the prohibition of accepting the unleavened breads of the Jews. We can see here that the Apostolic Canon and the Canons of Laodicea are very similar, despite the fact that they come from different regions of the Empire. This emphasizes the fact that Judaizing influences were very strong across the Eastern part of the Empire, especially those regarding Jewish Passover and Jewish traditions for Passover traditions.

The prohibition of participating in non-Orthodox feasts can be found in the canons of the Church not just regarding Jews but pagans or heretics as well. For example, Canon 39 of the Council of Laodicea forbids any attendance at pagan feasts and any communion "with their atheism (καὶ κοινωνεῖν τῇ ἀθεότητι αὐτῶν)". Canon 60 of the Council of Carthage forbade Christians to participate in pagan feasts because they involved lascivious dances, even on the same day as the commemoration of the martyrs (Perisanidi 2014, pp. 191–92).

The next canon referring to Jewish religious influences on Christians is the 62nd Apostolic canon, which states:

"If any Clergyman, for fear of human being, whether the latter be a Jew or a Greek or a heretic, should deny the name of Christ, let him be cast out and rejected; or if he denies the name of clergyman, let him be deposed; and if he repents, let him he accepted as a layman" (Cummings 1957, p. 107).

Despite their having been compiled in the mid-4th century, the Apostolic Canons often present the realities of previous Christian centuries, marked by an underdeveloped Church structure and administration, as a result of which the focus is limited to local or regional ecclesiastical development. The same reality can be found here. Under certain historical conditions, some Christians, even clergymen, were denying the name of Christ and thereby Christianity itself, becoming "apostates" (Milaș 1931, p. 280; Dron 1933, pp. 97–98). The punishment from the canons is clear and distinct for laymen and clergymen. In patristic literature, the Greek verb "ἀρνέομαι" is often used for apostatizing, denying a religious belief or religious identity (Lampe 1995, p. 227; Linder 2012, p. 197). Here, the abnegation or renunciation of the name of Christ through fear of those who do not share one's belief is considered a great sin, the apostate is excommunicated, and the clergyman who denies his sacerdotal identity is defrocked.

As can be observed here, the Jews were considered to be in the same group as pagans and heretics. If this canon refers to apostasy in times of persecution, then all three groups represented in different ways the persecutors of Christian clergy. The same kind of apostasy is condemned by St. Gregory of Nyssa in his second canon (Hosang 2010, p. 110). As in the Apostolic Canon, St. Gregory considers apostasy and conversion to Judaism as idolatry, heresy, and atheism.

Apostolic Canon 62 can similarly be considered evidence for the anti-Christian campaign of Emperor Julian in his short reign from 361–363. During his stay in Antioch, the Emperor started a persecution of Christians by blaming them for the destruction of the Temple of Apollo in Daphne. In order to diminish Christian influence, the Emperor supported the Jews by promising to rebuild their Temple (Hosang 2010, p. 112).

The 64th Apostolic Canon forbids clergymen, as well as laymen, to pray with Jews or heretics in their congregations. The same prohibition is stipulated in Canon 45, but here, the Jewish synagogue is specifically added, and the canon again applies to laymen. In both canons, the punishment is defrocking for clergymen and excommunication for laymen. It is known that at Antioch at the end of the 4th century there were two synagogues, one at Antioch and one in Daphne, a suburb of Antioch (Hosang 2010, p. 110).

Similar to Canon 64, Canon 71 forbids Christians from bringing offerings to a liturgical sacrifice in a pagan temple or a Jewish synagogue. It is probable that this series of canons regarding the prohibition of any liturgical or devotional con-celebration between Christians and Jews is a result of the fact that some Christians, attracted to Judaism, pretended to be "Sabbath goyim", i.e., persons who performed some tasks forbidden to Jews on the Sabbath and holidays (Feldman 1996, p. 376).

It can be observed in the Homilies of John Chrysostom that the canons had a low impact on the life of the Church: Christians were still respecting and celebrating Jewish feasts and fasting periods and were still attending synagogues.

As in the Apostolic Canons, John Chrysostom considers that the Jews can be compared to heretics because they share the same doctrinal errors regarding Christology, rejecting the divinity of Christ: "But the danger from this sickness is very closely related to the danger from the other; since the Anomians impiety is akin to that of the Jews, my present conflict is akin to my former one. And there is a kinship because the Jews and the Anomians make the same accusation" (Chrysostom 2010, p. 4).

## 3. Canons Regarding Judaizing Christians

Some of the prohibitions regarding Judaizing practices or Jewish liturgical influences on Christians can be found in the Canons of the Council of Laodicea. The Council of Laodicea, which can be dated between 338 and 381 (Ohme 2012, pp. 47–49; Huttner 2013,

pp. 290–96; Menevisoglu 1984, pp. 861–74), issued a number of canons in order to put an end to Judaizing practices found in the Christian communities.

Canon 16 of the Council, which apparently has nothing to do with this problem, imposes the requirement for reading the Gospels on Saturday together with the other biblical texts. Regarding this canon, three interpretations are possible, as emphasized by Hosang (Hosang 2010, p. 94). The first one underlines the fact that the Fathers of the Church imposed such a canonical rule in order to prevent any Jewish influence in church services on Saturdays (Schrechenberg 1990, p. 277; Trebilco 1991, p. 101). The Gospels, being the biblical texts that teach us about the redemptive activity of Christ and his resurrection, should prevail over other Old Testament texts. "The imposition of the Gospel reading on Saturday is considered to be due to Judaizing tendencies, with some Christians using exclusively Old Testament texts on Saturdays" (Schrechenberg 1990, p. 277). The second interpretation of this canon emphasizes the fact that some Christians were keeping the Jewish prayer service in the church on Saturday, but no consistent evidence can be found for such a practice. The third possible interpretation of this canon is given by the fact that some Christians were indeed attending Jewish synagogues on Saturdays (Trebilco 1991, p. 101; Hosang 2010, p. 93). It is known that the Jewish community had two synagogues in that region, one at Sardis, considered one of the biggest in the Jewish diaspora (Huttner 2013, p. 246; Kraabel et al. 1992, pp. 197–207), and one at Aphrodisias (Hosang 2010, p. 93).

The problem regarding the second and third interpretations of Canon 16 is that the authors overestimate the Jewish influence in Christian communities in the middle of the fourth century (Huttner 2013, pp. 67–80).

Canon 29 of the Council is aimed more directly against Judaizing practices: "Christians must not Judaize by resting on the Sabbath, but must work on that day, rather honouring the Lord's Day; and, if they can, resting then as Christians. But if any shall be found to be Judaizers, let them be anathema from Christ" (Percival 1900, p. 148). It is important to note that this canon (Hosang 2010, p. 93; Antic 2011, pp. 343–44) is the only one in the entire Canonical Collection that uses the verb "ἰουδαΐζω" and the noun "ἰουδαϊστής" (Menevisoglu 2013, p. 171). This can prove that the canons are directed at the Judaizing Christians. The verb "ἰουδαΐζω" is variously used in patristic literature for Christians who were embracing and practicing Judaism, for those who imitate the Jews by observing the Sabbath, or for heretics who deny the divinity of Christ (Lampe 1995, p. 674).

The canon imposes on Judaizing Christians one of the harshest canonical punishments, i.e., anathema, a total exclusion from the community. This highlights the fact that many Christians were still keeping the Sabbath as a day of rest and, probably, some other Jewish practices as well. This kind of religious syncretism, adopted by some Christians, is seen as a contradiction of Christian doctrine. The solution of the Fathers of the Council is to convert the Jewish Sabbath into a Christian working day (Trebilco 1991, p. 101; Hosang 2010, p. 96). Theophilus of Alexandria, in his first canon, addresses the same problem, emphasizing the superiority of Sunday over the Jewish Sabbath ("ἅτε δὴ ὑπεραναβεβηκυῖα τῶν Ἰουδαίων σαββατισμόν").

It is very important to take into account here the doctrinal context of the fourth century, dominated by the dispute regarding the divinity of Christ. As we saw above, the canons spoke of three groups: Jews, pagans, and heretics. The doctrinal error common to all these, as emphasized by the Church Fathers, is that they all rejected the divinity of Christ. On this account, the harsh canonical attitude of the Councils has to be seen as an instrument for rejecting any possible doctrinal influence of these groups on Christians, as well as any religious and liturgical influences of Jews and heretics on Christians through which these doctrines might find an open gateway into the liturgical life of the Church.

The canon is also enforced by a law promulgated by Constantine the Great on 3 March 321 that declared Sunday as a day of rest (Codex Justinianus III, 12.3). This law did not apply to the people from the countryside, which may be why the canon states that Christians should "as much as possible, refrain from work" on Sunday.

Other canons of the Council of Laodicea are considered to address Jewish influence on Christians. Canon 35 of the Council forbids the invocation of angels and attendance at distinct liturgical assemblies. Some authors connect this prohibition with Jewish influences. F. Hosang considers the veneration of angels to be a pagan influence and not a Jewish one (Hornung 2016, p. 193). Canon 36 of Laodicea condemns upper and lower clergy ("ἱερατικοὺς ἢ κληρικούς") who practice the witchcraft of magicians, enchanters, and astrologers, as well as the fabrication and usage of amulets. Some scholars also see Jewish practices here (Hosang 2010, p. 104), especially the use of amulets and incantations (Hornung 2016, p. 194; Lacerenza 2002, pp. 395–97).

The Canon of Patriarch Tarasios of Constantinople, written after the Seventh Ecumenical Council to Pope Hadrian I, condemns simony. Those who commit simony are considered "very much like Judas the traitor, who sold and betrayed Christ to the God-slaying Jews in exchange for pieces of silver" (Cummings 1957, p. 954). Even though Tarasius considered the Jews as "τοίς θεοκτόνοις Ἰουδαίοις", the phrase is used just for Judas and the Jews who received the money and does not imply a collective, multigenerational condemnation of Jews or a collective guilt of all Jews. This was a common rhetorical topos in Patristic Greek and Syriac literature (Soffer 2012, p. 855).

## 4. Canons Regarding the Prohibition of Jewish Social and Religious Influences on Christians

### 4.1. Marriages between Christians and Jews

One of the most important canons concerning mixed marriage in the fifth century is Canon 14 of the Fourth Ecumenical Council. Taking into consideration that the Churches of the East and West accepted this Council as the fourth Ecumenical one, its canonical provisions were considered normative for the entire Church.

As we can observe from its text, Canon 14 of the Council of Chalcedon (Tanner 1990, p. 94) reaffirms all the previous canonical decisions regarding mixed marriage. It is improbable that the Fathers of the Church were influenced by the Canon of Carthage, but what is more likely is that they knew the decisions of the Council of Laodicea, given that its decisions were part of the codex of canons used by the Fathers at the Fourth Ecumenical Council (Schwartz 1965, p. 95; Mardirossian 2010, p. 42).

Canon 14 of the Fourth Ecumenical Council represents evidence of the fact that previous canons of the Local Councils were not applied with strictness in the Church and members of the laity and lower clergy married "heterodox women". As we can see, the canon speaks about marriages with heretics and about children that have been baptized by heretics, forbidding children of Orthodox believers from giving their children in marriage to heretics, Jews, or pagans. If we analyze the Byzantine state legislation from this period, we can see that marriage with Jews or pagans was not a real problem that had to be solved by the Council (Troianos 1983, p. 98). Emperors Valentinian, Theodosius, and Arcadius had already forbidden any marriage between Jews and Christians. The law given at Thessalonica on 30 April (388) states that: "No Jew shall marry a Christian woman, nor a Christian man a Jewess. And if anyone does anything of the kind, the act shall be considered to be of the nature of adultery, and liberty of accusation is given to everyone" (Pharr 1952, p. 70).

As in the canons of the Council of Laodicea, Canon 14 of the Fourth Ecumenical Council linked marriage with baptism; the heretic spouse has to promise that he or she will convert to Orthodoxy. Zonaras'interpretation is very important for the meaning of this passage when he says that: "And if the baptism has not yet been done, let them not be baptized by heretics, nor in the name of their children unite them with the heretics, neither with the Jews, nor with the pagans. 'Heretics' are defined here as those that receive our Sacrament but who are mistaken in something, and are in disagreement with the Orthodox, 'Jews' are those who have killed Christ, and 'pagans' are those who are totally unbelievers and those who are worshiping idols" (Viscuso 1995, p. 236).

Theodor Balsamon divides heretics into two categories, "mistaken" heretics and "unfaithful" heretics, considering Jews and pagans as "unfaithful" heretics: "For you have seen that heretics are divided into two categories, into those that received our Mystery and the divine condescension, but who are mistaken in some things, and when they come to us, we anoint them only with myrrh, and into those that absolutely do not receive this, who are unfaithful, i.e., Jews and Greek, whom we also baptize" (Viscuso 1995, p. 240).

*4.2. Social Cohabitation between Christians and Jews*

Canon 11 of the Council of Trullo prohibits, inter alia, Church members from calling or visiting Jewish doctors in the case of sickness or receiving medical treatment from them, subjecting the clergy to defrocking and laymen to excommunication from the Eucharist (Nedungatt and Featherstone 1995, p. 81). This canon is often used as an argument for presupposed anti-Jewish or even anti-Semitic attitudes in the canons of the Church (Linder 2012, p. 200; Fishman-Duker 2012, p. 796). Some contemporary theologians see this text as having fallen into disuse, being unnecessary, a retrograde step, or even hilarious because the Christian would have to ask at the hospital if the doctor has any Jewish identity. On the other hand, some theologians consider the canon as being still normative in the Orthodox Church and applicable as such because "the present canon imposes this separatism" (Floca 1991, p. 105) between Jews and Christians. For these canonists, merely the existence of such a canonical decision represents an argument for considering it normative.

Indeed, the canon appears inapplicable in the socio-cultural context of our century because of its presupposed anti-Jewish canonical rule. That is why we are tempted to consider this text, along with many other patristic texts, as inapplicable and obsolete because they would imply social separatism between Christians and Jews. Some scholars consider this canon just "a rhetorical exercise in the juxtaposition of internal and external faith" (Toch 2012, p. 25). For this reason, a well-articulated contextual interpretation of the canon is necessary.

By this canon, the Fathers of the Council tried to ban that kind of medicine practiced by the Jews at the end of the seventh century, considered to be "iatromagia" (Ohme 2006, p. 67), a combination of Jewish ritual practices, amulets, and therapeutic techniques involving devotional practices. It is known that the Jews practiced this kind of medicine. The Fathers of the Council of Trullo condemned pseudo-medicine that involved the liturgical and devotional practices of any other religion. It is commonplace in patristic literature for some Jewish practices to be seen in connection to magic (Lacerenza 2002, pp. 395–97). Moreover, during that period, most Jewish doctors were rabbis, and in their healing acts, they used amulets, Old Testament verses as incantations, and other Jewish practices (Ohme 2006, pp. 66–67). The Fathers of the Council considered that such medical treatment with its implications for Jewish proselytism had to be banned. The same Council in Canon 61 explicitly condemns magic and divination, but these are considered "pagan" (Ohme 2006, pp. 48–57; Rochow 1978, p. 483–98; Fögen 1995, pp. 99–115).

Despite this canonical prohibition, Christians continued to visit Jewish physicians and pharmacists. At the beginning of the second millennium, Jewish physicians were still visited, even by the emperor (von Falkenhausen 2012, p. 884). This is proven by the fact that the Byzantine canonists of the 12th century saw this problem as a real one. In the 14th century, Patriarch Athanasius underlined this prohibition, as did, in the 15th century, the theologian Joseph Bryennius (Congourdeau 2012, pp. 716–17).

Canon 8 of the Seventh Ecumenical Council rejected false conversions of Jews to Christianity. Beyond the social and political context of that period that could have influenced these decisions, this canon represents a modern perspective on freedom of religion. As Father Grigorios Papathomas says: "this canon claims the freedom of religious expression and practice of hetero-religious persons and regulates the religious rights of Jews within society, specifically their freedom to remain Jews" (Papathomas 2013, p. 33). Although the civil Byzantine legislation tried to impose forced conversion of Jews to Christianity, the Seventh Ecumenical Council rejected such a practice.

## 5. Conclusions

As we saw, the canons of the Orthodox Church apply only to members of the Church: clergymen, laymen, and monastics. Accordingly, the canons found in the Canonical Collection influenced only indirectly the status of Jews in the Byzantine Empire by imposing a number of restrictions or prohibitions on Christians with regard to Jewish religious influences, Judaizing Church practices, and a Jewish–Christian religious syncretism, along with social interaction with Jews involving religious influences. All these prohibitions are based on doctrine. Out of a total of 770 canons in the Corpus canonum, only 21 canons refer to the Jews, in most cases only tangentially or together with other groups, such as heretics and pagans. This highlights the fact that there was no systematic concern of the canonical discipline of the Orthodox Church in regulating religious, social, economic, and cultural relations with Jews over time. Of the 20 canons, most of them (12) refer to prohibitions of Jewish religious influences on Christians, five canons refer to Judaizing practices or Jewish liturgical influences on Christians, and the other four canons refer to social relations with Jews. On the basis of the IHRA definition of anti-Semitism, we can conclude that the canonical material analyzed in this research does not represent expressions of hatred towards Jews. The letter or canon of Patriarch Tarasius of Constantinople contains a phrase against the Jews who killed Jesus Christ, but this phrase cannot be accepted as rhetorical manifestations directed toward Jews or their property. Despite the fact that some canons prohibited Christians from attending Jewish synagogues, they must not be considered manifestations toward Jewish community institutions and their religious facilities.

Seen in their historical, social, and cultural contexts, these canons represent solutions for diminishing a sort of Christian sympathy towards Jewish religious services and practices. Nonetheless, the most important feast of the Orthodox Church still takes into account the computation of Jewish Passover in order to calculate the Pascha in the Orthodox Church today. Bearing all this in mind, and following Robert Wilken, it would be incorrect to understand these canons, as did many twentieth-century historians, as intentionally anti-Semitic. It is important to examine in future academic research the theological, political, and cultural sources of the interpretation of the canons in the 20th century by Orthodox canonists as arguments to justify anti-Semitism in the context of the exalted manifestation of Nazi and fascist ideologies in Eastern Europe in the Inter-War Period.

**Funding:** This research received no external funding.

**Institutional Review Board Statement:** Not applicable.

**Informed Consent Statement:** Not applicable.

**Data Availability Statement:** Not applicable.

**Conflicts of Interest:** The authors declare no conflict of interest.

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
