# Peer review of "The Image of Jews According to the Canonical Tradition of the Orthodox Church"

_religions, doi:10.3390/rel14010091_

Round 1
Reviewer 1 Report
The topic addressed, as well as its analysis, challenges us all to an evaluation regarding the updating of the interpretation of the canons of the Fundamental Collection.
No suggestion!
Congratulations
Author Response
Dear Reviewer,
Many thanks for your observations. The language and the style will be rechecked and the text will be reviewed by a native English-speaking Professor.
Best regards and many thanks for your review.
Reviewer 2 Report
This paper analyzes canonical texts of Orthodox Christian (Autocephalous) Churches for anti-Semitic themes. The paper has the potential to offer an important contribution to the field but has several major gaps that must be addressed before it can realize its potential. For continued consideration, this manuscript will require extensive revisions. In no particular order, the gaps are as follows.
1. There is no treatment of research design considerations in this manuscript. Some MDPI articles have a Materials and Methods section. I am not insisting on a separate section for this manuscript, but 2-3 paragraphs addressing research design issues are needed. (a) Since this is a textual analysis, how were the analyzed texts sampled or selected from the larger universe of canonical texts if sampling was conducted? Regardless, a logical framework for textual selection must be specified. (b) Relatedly, are their judgment calls about canonical versus non-canonical texts, and could the raising of anti-Semitic texts to canonical status tell us something important about the textual "canonization" process and the historical transmission of anti-Semitism? This issue is lurking in the background but should be made explicit. (c) How were references to Jews, Jewishness, and Judaism determined by the author during the process of data analysis? Was a database on canonical texts searched and, if so, what search terms were used to identify relevant passages? (d) How were interpretations rendered? Was there a working definition of anti-Semitism that guided the analyses and interpretations? Were any modifications needed to that definition to keep it viable throughout the analysis? The research methodology is really missing here.
2. Some theoretical context would be useful here. A nod toward critical race theory (Jewishness as a social construct and a maligned racial-ethnic category at that) seems in order. Moreover, there has been good work on interpretive communities. Texts don't speak plainly or in a self-evident fashion. Meanings are imputed and affixed to them by groups of people collectively situated in a historical and cultural context. The practice of generating specific meanings from texts is lent credence from the analyses presented here. Since there is some attention to commentary on these canonical texts, the interpretive community perspective is an important backdrop.
3. Analyses of texts often highlight the contradictions at play within texts (pro- and anti-Jewish sentiments) as well as competing interpretations of any particular text (see, e.g., https://www.mdpi.com/2075-4698/8/3/55). Is there any evidence of such factors at work here? Such an angle would enrich the analysis.
4. How is anti-Semitism defined and is that definition rooted in current scholarship? I ask because anti-Semitism is often viewed as a continuum rather than a binary pro/anti category among researchers. So, there are different degrees of anti-Semitism, all deplorable but some more egregious than others. Moreover, there are different types of anti-Semitism. Some conservative religious groups, for example, can be very pro-Israel because of their (literal) interpretation of the Bible while still harboring much prejudice against Jews as a racial-ethnic group. A nod could be made to this research, and perhaps some of it can inform the results presented in the paper.
5. The Conclusions section is very inadequate. The author concludes that these Orthodox Christian practices do not imply anti-Semitism. Can things be really boiled down so definitively? In fact, the paper’s primary argument is undermined by such an assertion, as if to say “The analysis has been completed and there’s nothing of substance to see here.” A conclusion like that means that paper may not need to be published. Doesn’t this depend on how anti-Semitism is defined? Also, it seems that there is a mix of overt and covert forms of anti-Semitism (back to a continuum and different manifestations of anti-Semitism). A complicated story line seems to be evident here, but the concluding section gives no credence to complexity and undercuts the value of the analysis. The concluding section could also address directions for future research (ethnography and qualitative interviews examining interpretive practices come immediately to mind).
6. There are lots of typos in this manuscript. It needs careful proofing from an English speaker. Even the first paragraph uses legislator rather than legislature, but I found many errors throughout.
I wish the author well on these revisions, which are necessary and extensive.
Author Response
Dear Reviewer,
Many thanks for your outstanding remarks regarding my paper and thank you for taking the time to analyze the text and review it. I totally agree with your perspective, that can enrich the contribution in the field of Canon Law. Based on your review, I have tried to include the following changes in the final text:
- I have written about a page more on the analysis of the materials and methods used in the study in accordance with MDPI general methodology. I have dealt with the question of the choice of materials, the difference between canonical and non-canonical materials, how to select canons that relate to the image of the Jews in the Orthodox Canonical Collection. I have described the quantitative and qualitative analysis used in the research.
The following part was included:
„ The initial research material for the present study consisted of the Collection of Canons of the Orthodox Church, approximately 770 canons (Wagschal 2015, pp. 60-61), which are still considered normative for the ecclesial organization and church discipline by all Autocephalous Orthodox Churches. The research material did not include non-canonical texts from collections that were not recognized by the Eastern Orthodox Church in her Corpus Canonum (Troianos 2012, pp. 115-124), due to their local usage and non-obligatory canonical status. The first stage of the research was to identify from these 770 canons those texts that imposed, as we will see, some restrictions or prohibitions on Christians regarding Jewish religious influences, Judaize Church practices or a Jew-ish-Christian religious syncretism, and social and religious interaction with Jews. The main material used for the present research consisted of 21 canons from the Canonical Collection of the Orthodox Church. This material represents all the canons from the Corpus Canonum regarding Jews. The selected material was grouped into three categories, to which I have added subcategories. The main purpose of this thematic content analysis was to identify thematic categories of canonical restrictions regarding Jews in order to compare them with thematic categories of anti-Semitism. The second step is to see the presence or absence of certain categories by applying quantitative content analysis. It is really im-portant to emphasize from the start the difference between the legal provisions regarding Jews in the Byzantine period (Linder 2006, pp. 128–169), the so-called Jewry Law (Linder 2012, pp. 195–206) that had a direct influence on the legal status of Jews in society and Canon Law applied only to members of the Church: clergymen, laymen, and monastics. Accordingly, the canons found in the Canonical collection influenced only indirectly the status of Jews in the Byzantine Empire. This research uses content analysis as the main method applied to the canonical material, based on the principles of the interpretation of the canons and legal texts in the Orthodox Church (PerÈ™a 2021, pp. 445-547). The selected canonical material was analyzed using both quantitative and qualitative methods, based on current research on religious studies (Brink 1995, pp. 461-475) and legal or canonical texts (Stanisz 2022, p. 5). From the perspective of qualitative content analysis, the ca-nonical material was subjected to synchronic and diachronic interpretation in order to emphasize the reception history or Wirkungsgeschichte of these texts in the Tradition of the Orthodox Church.”
- From the point of view of Canon Law and Orthodox Canon Law, it is very difficult to link the topic with Critical race theory for several reasons. The application of CRT in the field of Canon Law and Ancient Christianity is still under development. In his paper 2009, „Be Not Afraid of the Dark: Critical Race Theory and Classical Studies”, Shelley P. Haley emphasizes the possibility of such an approach. In 2010, Eric D. Barreto, in the introduction of his book: „Ethnic Negotiations: The Function of Race and Ethnicity in Acts 16” shows different positions of current academic scholars regarding the possibility and limits of the application of CRT in studies regarding Ancient Christianity. From this point of view, I tried to emphasize the fact that for the canons the Jewishness is considered more a religious and doctrinal group, and not a racial or ethnical group, as in the Jewry-Law of the Byzantine period.
Instead of interpretative community perspective, I have used the method of Wirkungsgeschichte or „The history of effect/reception”, according to the theory of Hans Georg Gadamer. This is emphasized in the new chapter regarding materials and methods.
- I the text I tried to emphasize the pro and anti-Jewish attitudes of the Christian believers during time. The multitude of canonical prohibitions emphasizes the fact that Christians were still using Jewish religious practices and were influenced by Jewish religious habits.
- I tried to find an accepted definition of anti-Semitism and anti-Judaism. Initially I avoided this because there is no unanimously accepted definition of anti-Semitism. The working-definition is emphasized on the base of current research. In order to see if the Corpus Canonum contains forms of anti-Semitism or exclusion of Jews on the basis of these texts, a working definition of anti-Semitism was needed. The difficulty of such a definition is that no attempt to define antisemitism has been unanimously accepted by current academic research. Some scholars impose a difference between anti-Semitism, as racial discrimination towards Jews in modern times and anti-Judaism, as form of rational, nonrational, and irrational response to Jewish religious influences (G. Langmuir 1990, pp. 57-99). This distinction was criticized by many scholars (Schäfer 1997, pp. 197-211), but this pioneered direction was developed by further academic research as it was emphasized by Robert Chazan in his recent book (Chazan 2016, pp. viii-xi).
- I have changed the Conclusion section entirely. I have included address directions for future research.
- The text was reviewed by an English speaker Professor of Theology and I am currently introducing the corrections. The final text will be uploaded today.
Round 2
Reviewer 2 Report
I commend the author on a sound and comprehensive revision. I would quibble with the conclusion that the canons examined here lack anti-Jewish references or sentiments. Efforts to distinguish Christianity from Judaism may not be inherently anti-Semitic. So, the initial message and authorial intent may not be intrinsically racist. However, as cultural artifacts, canonical texts could later be used by those with racist intentions to denigrate Jews and Judaism, could they not? There is research on overt versus covert forms of racism, and it is possible that statements whose authors had no racist intent could later be leveraged by racists to foster anti-Semitism. This point relates to debates about how meaning(s) are invested in texts. Authorial intent is one form of meaning, but readers (even centuries later) with anti-Jewish standpoints could impute alternative meanings that aim to move the textual passages in more anti-Semitic directions. Could a point like this not be conceded in the Discussion section? It may indeed require additional research, but faith traditions are living entities guided by texts that are polysemous. Thus, a less definitive conclusion seems warranted.
Author Response
Dear Reviewer,
Many thanks for your remarks regarding my paper and thank you for taking the time to reanalyse the text.
On the basis of your review, a have made the following changes:
I have rechecked the minor spelling error in the text. I have checked the arguments and the structure of the argumentation in order to present the results according to empirical research and the hypotheses and methodology from the first chapter. The IHRA definition of Anti-Semitism is the working definition accepted by academic, political and governmental bodies. Your review has been a great help in refocusing my academic study. As I have concluded, according to this working definition the canons do not represent expressions of hatred towards Jews.
The goal of the paper was to see if the canons are Anti-Semitic expressions. In the methodological section (rows 100-128) I have stated the difference between Anti-Jewish and Anti-Semitic attitudes, according to the thesis of G. Langmuir and Chazan. The purpose of the paper (row 10) was to identify if the canons are Anti-Semitic. I have limited my paper to this goal. As I stated in rows 18-64, in the context of the manifest exaltation of Nazi and fascist ideologies in Romania and Eastern Europe in the inter-war period, we had some Orthodox Theologians who considered the canons as arguments against the Jewish community. I totally agree with you that canonical texts could later be used by those with racist intentions to denigrate Jews and Judaism, but I tried to limit my paper to the official and canonical position of the Orthodox Church expressed in the canons, not going further to particular theologians of the 20th century. It would have been impossible to analyse the historical, cultural, ideological, and canonical context of such positions of those theologians in the same paper. This issue will be analysed in a future paper, as I said in the final part of my research: „It is important to examine in future academic research the theological, political and cultural sources of the interpretation of the canons in the 20th century by Orthodox canonists as arguments to justify anti-Semitism in the context of the exalted manifestation of Nazi and fascist ideologies in Eastern Europe in the Inter-War Period”.
Once again,
Many thanks for your review.